# End-to-End Weak Supervision

**Salva Rühling Cachay**[1,2*]  **Benedikt Boecking**[1]  **Artur Dubrawski**[1]

[1] Carnegie Mellon University  [2] Technical University of Darmstadt

## Abstract

Aggregating multiple sources of weak supervision (WS) can ease the data-labeling bottleneck prevalent in many machine learning applications, by replacing the tedious manual collection of ground truth labels. Current state of the art approaches that do not use any labeled training data, however, require two separate modeling steps: Learning a probabilistic latent variable model based on the WS sources – making assumptions that rarely hold in practice – followed by downstream model training. Importantly, the first step of modeling does not consider the performance of the downstream model. To address these caveats we propose an end-to-end approach for directly learning the downstream model by maximizing its agreement with probabilistic labels generated by reparameterizing prior probabilistic posteriors with a neural network. Our results show improved performance over prior work in terms of end model performance on downstream test sets, as well as in terms of improved robustness to dependencies among weak supervision sources.

## 1 Introduction

The success of supervised machine learning methods relies on the availability of large amounts of labeled data. The common process of manual data annotation by humans, especially when domain experts need to be involved, is expensive, both in terms of effort and cost, and as such presents a major bottleneck for deploying supervised learning methods to new domains and applications.

Recently, data programming, a paradigm that makes use of multiple sources of noisy labels, has emerged as a promising alternative to manual data annotation [30]. It encompasses previous paradigms such as distant supervision from external knowledge bases [29, 34], crowdsourcing [15, 26, 14, 42], and general heuristic and rule-based labeling of data [23, 20]. In the data programming framework, users encode domain knowledge into so called labeling functions (LFs), which are functions (e.g. domain heuristics or knowledge base derived rules) that noisily label subsets of data. The main task for learning from multiple sources of weak supervision is then to recover the sources' accuracies in order to estimate the latent true label, without access to ground truth data. In previous work [30, 32, 19], this is achieved by first learning a generative probabilistic graphical model (PGM) over the weak supervision sources and the latent true label to estimate *probabilistic labels*, which are then used in the second step to train a *downstream model* via a noise-aware loss function.

Data programming has led to a wide variety of success stories in domains such as healthcare [18, 17] and e-commerce [5], but the existing PGM based frameworks still come with a number of drawbacks. The separate PGM does not take the predictions of the downstream model into account, and indeed this model is trained independently of the PGM. In addition, current approaches for estimating the unknown class label via a PGM need to rely on computationally expensive approximate sampling methods [30], estimation of the full inverse of the LFs covariance matrix [32], or they need to make strong independence assumptions [19]. Furthermore, existing prior work and the associated theoretical analyses make assumptions that may not hold in practice [30, 32, 19], such as availability

---

*salvaruehling@gmail.com.

35th Conference on Neural Information Processing Systems (NeurIPS 2021).

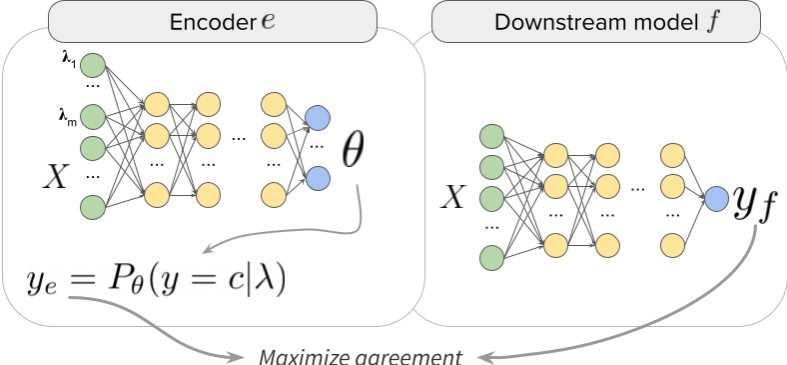

Figure 1: For a task with unobserved ground truth labels $y$, given $m$ sources of weak supervision $\lambda_i$ and training features $X$, WeaSEL trains a downstream model $f$ by maximizing the agreement of its predictions $y_f$ with probabilistic labels $y_e = P_\theta(y = c | \boldsymbol{\lambda})$ generated by reparameterizing the posterior of prior work with sample-dependent accuracy scores $\theta$ produced by an encoder network $e$.

of a well-specified generative model structure (i.e. that the dependencies and correlations between the weak sources have been correctly specified by the user), that LF errors are randomly distributed across samples, and that the latent label is independent of the features given the weak labels (i.e. only the joint distribution between the sources and labels needs to be modeled).

We introduce WeaSEL, our Weakly Supervised End-to-end Learner model for training neural networks with, exclusively, multiple sources of weak supervision as noisy signals for the latent labels. WeaSEL is based on 1) *reparameterizing previous PGM based posteriors* with a neural encoder network that produces *accuracy scores for each weak supervision source*; and 2) training the encoder and downstream model on *the same target loss, using the other model's predictions as constant targets*, to maximize the agreement between both models. The proposed method needs no labeled training data, and neither assumes sample-independent source accuracies nor redundant features for latent label modeling. We show empirically that it is not susceptible to highly correlated LFs. In addition, the proposed approach can learn from multiple probabilistic sources of weak supervision.

Our contributions include:

- We introduce a flexible, end-to-end method for learning models from multiple sources of weak supervision.
- We empirically demonstrate that the method is naturally robust to adversarial sources as well as highly correlated weak supervision sources.
- We release an open-source, end-to-end system for arbitrary PyTorch downstream models that will allow practitioners to take advantage of our approach[2].
- We show that our method outperforms, by as much as 6.1 F1 points, state-of-the-art latent label modeling approaches on 4 out of 5 relevant benchmark datasets, and achieves state-of-the-art performance on a crowdsourcing dataset against methods specifically designed for this setting.

## 2 Related Work

**Multi-source Weak Supervision**   The data programming paradigm [30] allows users to programmatically label data through multiple noisy sources of labels, by treating the true label as a latent variable of a generative PGM. Several approaches for learning the parameters of the generative model have been introduced [32, 19, 11] to address computational complexity issues. Existing methods are susceptible to misspecification of the dependencies and correlations between the LFs, which can lead to substantial losses in performance [8]. Indeed, it is common practice to assume a conditionally independent model – without any dependencies between the sources – in popular libraries [5, 31] and related research [15, 2, 37, 7], even though methods to learn the intra-LF structure have been proposed [4, 39, 38]. In natural language processing, [40] consider more relaxed and broader forms

---

[2]https://github.com/autonlab/weasel

of weak supervision and introduce a framework that uses virtual evidence as prior belief over latent labels and their interdependencies, learning an end model jointly with the label model via variational EM. As in the approach proposed in this paper, the aforementioned methods do not assume any labeled training data, i.e. the downstream model is learned based solely on outputs of multiple LFs on unlabeled data. The traditional co-training paradigm [6] on the other hand is similar in spirit but requires some labeled data to be available. Recent methods that study the co-training setup where labeled training data supplements multiple WS sources, include [3, 25]. Note that the experiments in [3, 25] rely on large pre-trained language models, making the applicability of the approach without such models or to non-text domains unclear.

**Crowdsourcing** Aggregating multiple noisy labels is also a core problem studied in the crowd-sourcing literature. Common approaches model worker performance and the unknown label jointly [15, 14, 42] using expectation maximization (EM) or similar approaches. Some core differences to learning from weak supervision sources are that errors by crowdworkers are usually assumed to be random, and that task assignment is not always fixed but can be optimized for. The benefits of jointly optimizing the downstream model and the aggregator of the weak sources have been recognized in multiple end-to-end methods that have been proposed for the crowdsourcing problem setting [33, 22, 41, 27, 35, 9]. They often focus on image labeling and EM-like algorithms for modeling and aggregating the workers. Importantly, our proposed approach can be used in general applications with weak supervision from multiple sources without any restrictive assumptions specific to crowdsourcing, and we show that our approach outperforms the aforementioned methods on a crowdsourcing benchmark task.

# 3 End-to-End Weak Supervision

In this section we present our flexible base algorithm that we call WeaSEL, which can be extended to probabilistic sources and other network architectures (Section 7). See Algorithm 1 for its pseudocode.

## 3.1 Problem Setup

Let $(\mathbf{x}, y) \sim \mathcal{D}$ be the data generating distribution, where the unknown labels belong to one of $C$ classes: $y \in \mathcal{Y} = \{1, ..., C\}$. As in [30], users provide an unlabeled training set $\mathbf{X} = \{\mathbf{x}_i\}_{i=1}^N$, and $m$ labeling functions $\boldsymbol{\lambda} = \boldsymbol{\lambda}(\mathbf{x}) \in \{0, 1, ..., C\}^m$, where 0 means that the LF abstained from labeling for any class. We write $\bar{\boldsymbol{\lambda}} = (\mathbb{1}\{\boldsymbol{\lambda} = 1\}, \ldots, \mathbb{1}\{\boldsymbol{\lambda} = C\}) \in \{0, 1\}^{m \times C}$ for the one-hot representation of the LF votes provided by the $m$ LFs for $C$ classes. Our goal is to train a downstream model $f : \mathcal{X} \to \mathcal{Y}$ on a *noise-aware* loss $L(y_f, y_e)$ that

---

**Algorithm 1**  WeaSEL: The proposed Weakly Supervised End-to-end Learning algorithm for learning from multiple weak supervision sources.

---

**input:** batch size $n$, networks $e$, $f$, inverse temperatures $\tau_1, \tau_2$, noise-aware loss function $L$, class balance $P(y)$.
**for** sampled minibatch $\{z^{(k)} = (\mathbf{x}^{(k)}, \boldsymbol{\lambda}^{(k)})\}_{k=1}^n$ **do**
  **for all** $k \in \{1, \ldots, n\}$ **do**
    # Produce accuracy scores for all weak sources
    $\theta\left(z^{(k)}\right) = \mathrm{softmax}\left(e(z^{(k)})\tau_1\right)$
    # Generate probabilistic labels
    **define** $\mathbf{s}^{(k)}$ **as** $\mathbf{s}^{(k)} = \theta(z^{(k)})^T \bar{\boldsymbol{\lambda}}^{(k)}$
    $y_e^{(k)} = P_\theta(y|\boldsymbol{\lambda}^{(k)}) = \mathrm{softmax}\left(\mathbf{s}^{(k)}\tau_2\right) \odot P(y)$
    # Downstream model forward pass
    $y_f^{(k)} = f(\mathbf{x}^{(k)})$
  **end for**
  $\mathcal{L}_f = \frac{1}{n}\sum_{k=1}^n L\left(y_f^{(k)}, \texttt{stop-grad}\left(y_e^{(k)}\right)\right)$
  $\mathcal{L}_e = \frac{1}{n}\sum_{k=1}^n L\left(y_e^{(k)}, \texttt{stop-grad}\left(y_f^{(k)}\right)\right)$
  update $e$ to minimize $\mathcal{L}_e$, and $f$ to minimize $\mathcal{L}_f$
**end for**
**return** downstream network $f(\cdot)$

---

operates on the model's predictions $y_f = f(\mathbf{x})$ and *probabilistic labels* $y_e$ generated by an encoder model $e$ that has access to LF votes, $\boldsymbol{\lambda}$, and features, $\mathbf{x}$. Note that prior work restricts the probabilistic labels to only being estimated from the LFs.

## 3.2 Posterior Reparameterization

Previous PGM based approaches assume that the joint distribution $p(\boldsymbol{\lambda}, y)$ of the LFs and the latent true label can be modeled as a Markov Random Field (MRF) with pairwise dependencies between weak supervision sources [30, 31, 32, 19, 11]. These models are parameterized by a set of LF accuracy and intra-LF correlation parameters and in some cases by additional parameters to model

LF and class label propensity. Note however, that the aforementioned models ignore features $X$ when modeling the latent labels and therefore disregard that LFs may differ in their accuracy across samples and data slices.

We relax these assumptions, and instead view the latent label as an *aggregation of the LF votes that is a function of the entire set of LF votes and features, on a sample-by-sample basis*. That is, we model the probability of a particular sample $\mathbf{x}$ having the class label $c \in \mathcal{Y}$ as

$$P_\theta(y = c \,|\, \boldsymbol{\lambda}) = \operatorname{softmax}(\mathbf{s})_c \, P(y = c), \tag{1}$$

$$\mathbf{s} = \theta(\boldsymbol{\lambda}, \mathbf{x})^T \bar{\boldsymbol{\lambda}} \in \mathbb{R}^C. \tag{2}$$

where $\theta(\boldsymbol{\lambda}, \mathbf{x}) \in \mathbb{R}^m$ weighs the LF votes on a sample-by-sample basis and the softmax for class $c$ on $s$ is defined as

$$\operatorname{softmax}(\mathbf{s})_c = \frac{\exp\left(\theta(\boldsymbol{\lambda}, \mathbf{x})^T \mathbb{1}\{\boldsymbol{\lambda} = c\}\right)}{\sum_{j=1}^C \exp\left(\theta(\boldsymbol{\lambda}, \mathbf{x})^T \mathbb{1}\{\boldsymbol{\lambda} = j\}\right)}.$$

While we do not use the class balance $P(y)$ in our experiments for our own model, `WeaSEL`, it is frequently assumed to be known [32, 19, 11], and can be estimated from a small validation set, or using LF outputs as described in [32]. Our formulation can be seen as a reparameterization of the posterior of the pairwise MRFs in [31, 32, 19], where $\theta$ corresponds to the LF accuracies that are fixed across the dataset and are solely learned via LF agreement and disagreement signals, ignoring the informative features. We further motivate this formulation and expand upon this connection in the appendix A.

### 3.3 Neural Encoder

Based on the setup introduced in the previous section and captured in Eq. (1), our goal is to estimate latent labels by means of learning sample-dependent accuracy scores $\theta(\boldsymbol{\lambda}, \mathbf{x})$, which we propose to parameterize by a neural encoder $e$. This network takes as input the features $\mathbf{x}$ and the corresponding LF outputs $\boldsymbol{\lambda}(\mathbf{x})$ for a data point, and outputs unnormalized scores $e(\boldsymbol{\lambda}, \mathbf{x}) \in \mathbb{R}^m$. Specifically, we define

$$\theta(\boldsymbol{\lambda}, \mathbf{x}) = \tau_2 \cdot \operatorname{softmax}(e(\boldsymbol{\lambda}, \mathbf{x}) \tau_1), \tag{3}$$

where $\tau_2$ is a constant factor that scales the final softmax transformation in relation to the number of LFs $m$, and is equivalent to an inverse temperature for the output softmax in Eq. 1. It is motivated by the fact that most LFs are sparse in practice, and especially when the number of LFs is large this leads to small accuracy magnitudes without scaling (since, without scaling, the accuracies after the softmax sum up to one)[3]. $\tau_1$ is an inverse temperature hyperparameter that controls the smoothness of the predicted accuracy scores: The lower $\tau_1$ is, the less emphasis is given to a small number of LFs – as $\tau_1 \to 0$, the model aggregates according to the equal weighted vote. The $\operatorname{softmax}$ transformation naturally encodes our understanding of wanting to aggregate the weak sources to generate the latent label.

### 3.4 Training the Encoder

The key question now is how to train $e$, i.e. how can we learn an accurate mapping of the sample-by-sample accuracies, given that we do not observe any labels?

First, note that initializing $e$ with random weights will lead to latent label estimates close to an equal weighted vote, which acts as a reasonable baseline for label models in data programming (and crowdsourcing), where in expectation votes of LFs are assumed to better than random guesses. Thus, $P_\theta(y \,|\, \boldsymbol{\lambda}, \mathbf{x})$ will provide a better than random initial guess for $y$. *We hypothesize that in most practical cases, features, latent label, and labeling function aggregations are intrinsically correlated due to the design decisions made by the users defining the features and LFs. Thus, we can jointly optimize $e$ and $f$ by maximizing their agreement with respect to the target downstream loss $L$ in an end-to-end manner.* See Algorithm 1 for pseudocode of the resulting `WeaSEL` algorithm. The natural classification loss is the cross-entropy, which we use in our experiments, but in order to encode our desire of maximizing the agreement of the two separate models that predict based on different views of the data, we adapt it[4] in the following form: The loss is symmetrized in order to compute the

---

[3]In our main experiments we set $\tau_2 = \sqrt{m}$.

[4]This holds for any asymmetric loss, while for symmetric losses this is not needed.

Table 1: The final test F1 performance of various multi-source weak supervision methods over seven runs, using different random seeds, are averaged out ± standard deviation. The top 2 performance scores are highlighted as **First**, **Second**. Triplet-median [11] is not listed as it only converged for IMDB with 12 LFs (F1 = 73.0 ± 0.22), and Spouse (F1 = 48.7 ± 1.0). The downstream model is the same for all methods. For Sup. (Val. set), and Majority vote it is trained on the hard labels induced by the labeled validation set and the majority vote of the LFs, respectively. For the rest it is trained on the probabilistic labels estimated by the respective state-of-the-art latent label model. For reference, we also report the *Ground truth* performance of the same downstream model trained on the true training labels (which are unused by all other models, and not available for Spouse).

| Model | Spouse (9 LFs) | ProfTeacher (99 LFs) | IMDB (136 LFs) | IMDB (12 LFs) | Amazon (175 LFs) |
|---|---|---|---|---|---|
| Ground truth | – | $90.65 \pm 0.29$ | $86.72 \pm 0.40$ | $86.72 \pm 0.40$ | $92.93 \pm 0.68$ |
| Sup. (Val. set) | $20.4 \pm 0.2$ | $73.34 \pm 0.00$ | $68.76 \pm 0.00$ | $68.76 \pm 0.00$ | $84.18 \pm 0.00$ |
| Snorkel | $48.79 \pm 2.69$ | $85.12 \pm 0.54$ | $\mathbf{82.22 \pm 0.18}$ | $\mathbf{74.45 \pm 0.58}$ | $80.54 \pm 0.41$ |
| Triplet | $45.88 \pm 3.64$ | $74.43 \pm 10.59$ | $75.36 \pm 1.92$ | $73.15 \pm 0.95$ | $75.44 \pm 3.21$ |
| Triplet-Mean | $\mathbf{49.94 \pm 1.47}$ | $82.58 \pm 0.32$ | $79.03 \pm 0.26$ | $73.18 \pm 0.23$ | $79.44 \pm 0.68$ |
| Majority vote | $40.67 \pm 2.01$ | $\mathbf{85.44 \pm 0.37}$ | $80.86 \pm 0.28$ | $74.13 \pm 0.31$ | $\mathbf{84.20 \pm 0.52}$ |
| WeaSEL | $\mathbf{51.98 \pm 1.60}$ | $\mathbf{86.98 \pm 0.45}$ | $82.10 \pm 0.45$ | $\mathbf{77.22 \pm 1.02}$ | $\mathbf{86.60 \pm 0.71}$ |

gradient of both models using the other model's predictions as targets. To that end, it is crucial to use the `stop-grad` operation on the targets (the second argument of $L$), i.e. to treat them as though they were ground truth labels. This choice is supported by our synthetic experiment and ablations. This operation has also been shown to be crucial in siamese, non-contrastive, self-supervised learning, both empirically [21, 12] and theoretically [36]. By minimizing simultaneously, both, $L(y_e, y_f)$ and $L(y_f, y_e)$ to jointly learn the network parameters for $e$ and the downstream model $f$ respectively, we learn the accuracies of the noisy sources $\boldsymbol{\lambda}$ that best explain the patterns observed in the data, and vice versa the feature-based predictions that are best explained by aggregations of LF voting patterns.

### 3.5 WeaSEL Design Choices

Note that it is necessary to encode the inductive bias that the unobserved ground truth label $y$ is a (normalized) linear combination of LF votes – weighted by sample- and feature-dependent accuracy scores. Otherwise, if the encoder network directly predicts $P_\theta(y | \boldsymbol{\lambda}, \mathbf{x})$ instead of the accuracies $\theta(\boldsymbol{\lambda}, \mathbf{x})$, the pair of networks $e, f$ have no incentive to output the desired latent label, without observed labels. We do acknowledge that this two-player cooperation game with strong inductive biases could still allow for degenerate solutions. However, we empirically show that our simple `WeaSEL` model that goes beyond multiple earlier WS assumptions is 1) competitive and frequently outperforms state-of-the-art PGM-based and crowdsourcing models (see Tables 1 and 2); and 2) is robust against massive LF correlations and able to recover the performance of a fully supervised model on a synthetic example, while all other models break in this setting (see section 4.3 and appendix F).

## 4 Experiments

**Datasets** As in related work on label models for weak supervision [30, 32, 19, 11], we focus for simplicity on the binary classification case with unobserved ground truth labels $y \in \{-1, 1\}$. See Table 3 for details about dataset sizes and the number of LFs used. We also run an experiment on a multi-class, crowdsourcing dataset (see subsection 4.2). We evaluate the proposed end-to-end system for learning a downstream model from multiple weak supervision sources on previously used benchmark datasets in weak supervision work [31, 7, 11]. Specifically, we evaluate test set performance on the following classification datasets:

- *The IMDB movie review* dataset [28] contains movie reviews to be classified into positive and negative sentiment. We run two separate experiments, where in one we use the same 12 labeling functions as in [11], and for the other we choose 136 text-pattern based LFs. More details on the LFs can be found in the appendix C.

- A subset of the *Amazon review* dataset [24], where the task is to classify product reviews into positive and negative sentiment.

- We use the *BiasBios biographies* dataset [16] to distinguish between binary categories of frequently occurring occupations and use the same subset of professor vs teacher classification as in [7].
- Finally, we use the highly unbalanced *Spouse* dataset (90% negative class labels), where the task is to identify mentions of spouse relationships amongst a set of news articles from the Signal Media Dataset [13].

For the Spouse dataset, the same data split and LFs as in [19] are used, while for the rest we take a small subset of the test set as validation set. This is common practice in the related work [31, 32, 19, 7] for tuning hyperparameters, and allows for a fair comparison of models.

## 4.1 Benchmarking Weak Supervision Label Models

To evaluate the proposed system, we benchmark it against state-of-the-art systems that aggregate multiple weak supervision sources for classification problems, without any labeled training data. We compare our proposed approach with the following systems: 1) *Snorkel*, a popular system proposed in [31, 32]; 2) *Triplet*, exploits a closed-form solution for binary classification under certain assumptions [19]; and 3) *Triplet-mean* and *Triplet-median* [11], which are follow-up methods based on *Triplet* with the aim of making the method more robust.

We report the held-out test set performance of WeaSEL's downstream model $f$. Note that in many settings it is often not possible to apply the encoder model to make predictions at test time, since the LFs usually do not cover all data points (e.g. in Spouse only 25.8% of training samples get at least one LF vote), and can be difficult to apply to new samples (e.g. when the LFs are crowdsourced annotations). In contrast, the downstream model is expected to generalize to arbitrary unseen data points.

We observe strong results for our model, with 4 out of 5 top scores, and a lift of 6.1 F1 points over the next best label model-based method in the Amazon dataset. Our results are summarized in Table 1. Since our model is based on a neural network, we hypothesize that the large relative lift in performance on the Amazon review dataset is due to it being the largest dataset size on which we evaluate on – we expect this lift to hold or become larger as the training set size increases. To obtain the comparisons shown in Table 1, we run Snorkel over six different label model hyperparameter configurations, and train the downstream model on the labels estimated by the label model with the best AUC score on the validation set. We do not report Triplet-median in the main table, since it only converged for the two tasks with very small numbers of labeling functions. Interestingly, we observed that training the downstream model on the hard labels induced by majority vote leads to a competitive performance, better than triplet methods in four out of five datasets. This baseline is not reported in previous papers (only the raw majority vote is usually reported, without training a classifier). Our own model, WeaSEL, on the other hand consistently improves over the majority vote baseline (which in Table 4, in the appendix, can be seen to lead to similar performance as an untrained encoder network, $e$, that is left at its random initialization).

## 4.2 Crowdsourcing dataset

Data programming and crowdsourcing methods have been rarely compared against each other, even though the problem setup is quite similar. Indeed, end-to-end systems specifically for crowdsourcing have been proposed [33, 27, 35, 9]. These methods follow crowdsourcing-specific assumptions and modeling choices (e.g. independent crowdworkers, a confusion matrix model for each worker, and in general build upon [15]). Still, since crowdworkers can be seen as a specific type of labeling functions, the performance of general WS methods on crowdsourcing datasets is of interest, but has so far not been studied. We therefore choose to also evaluate our method on the multi-class LabelMe image classification dataset that was previously used in the core related crowdsourcing literature

Table 2: Test accuracy scores on the crowd-sourced, multi-class LabelMe image classification dataset.

| Model | Accuracy |
|---|---|
| Majority vote | $79.23 \pm 0.5$ |
| MBEM [27] | $76.84 \pm 0.4$ |
| DoctorNet [22] | $81.31 \pm 0.4$ |
| CrowdLayer [35] | $82.83 \pm 0.4$ |
| AggNet [1] | $84.35 \pm 0.4$ |
| MaxMIG [9] | $\mathbf{85.45} \pm \mathbf{1.0}$ |
| Snorkel+CE | $82.89 \pm 0.7$ |
| WeaSEL+CE | $82.46 \pm 0.8$ |
| Snorkel+MIG | $85.15 \pm 0.8$ |
| WeaSEL+MIG | $\mathbf{86.36} \pm \mathbf{0.3}$ |

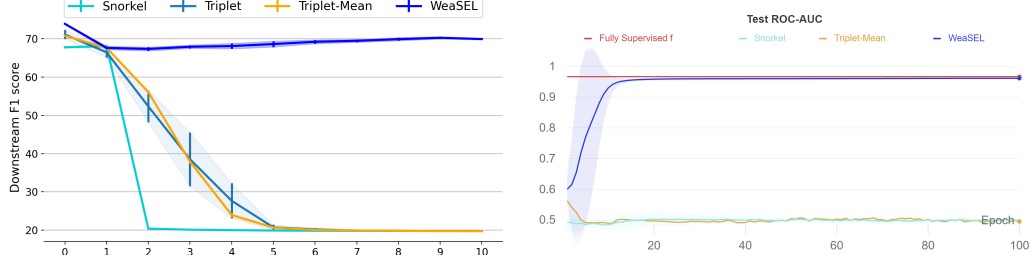

(a) Test F1 score on robustness experiment as a function of the number of adversarial LFs.

(b) Test AUC by epoch in an experiment where one LF corresponds to the true class label and others are random.

Figure 2: `WeaSEL` is significantly more robust against correlated adversarial (left) or random (right) LFs than prior work whose assumptions make them equivalent to a Naive Bayes model. For subfigure (a), we duplicate a fake adversarial LF up to 10 times, and observe that our end-to-end system is robust against the adversarial LF, while other systems quickly degrade in performance (over ten random seeds). In (b), we let one LF be the true labels $y^*$ and then duplicate a LF that votes according to a coin flip 2, 5, ..., 2000 times. We plot the test AUC performance curve as a function of the epochs, averaged out over the different number of duplications (and five random seeds). `WeaSEL` consistently recovers the test performance of the supervised end-model $f$ trained directly on the true labels $y^*$, whose end performance (AUC = 0.967) is shown in red.

[35, 9]. The results are reported in Table 2, and more details on this experiment can be found in Appendix E.

Note that the evaluation procedure in [9] reports the best test set performance for all models, while we follow the more standard practice of reporting results obtained by tuning based on a small validation set – as in our main experiments. We find that our model, `WeaSEL`, is able to outperform Snorkel as well as multiple state-of-the-art methods that were specifically designed for crowdsourcing (including several end-to-end approaches). Interestingly, this is achieved by using the mutual information gain loss (MIG) function introduced in [9], which significantly boosts performance of both Snorkel (the end-model, $f$, trained on the MIG loss with respect to soft labels generated by the first Snorkel label model step) and `WeaSEL` that use the cross-entropy (CE) loss. This suggests that the MIG loss is a great choice for the special case of crowdsourcing, due to its strong assumptions common to crowdsourcing which are much less likely to hold for general LFs. This is reflected in our ablations too, where using the MIG loss leads to a consistently worse performance on our main multi-source weak supervision datasets.

## 4.3 Robustness to Adversarial LFs and LF correlations

Users will sometimes generate sources they mistakenly think are accurate. This also encompasses the 'Spammer' crowdworker-type studied in the crowdsourcing literature. Therefore, it is desirable to build models that are robust against such sources. We argue that our system that is trained by maximizing the agreement between an aggregation of the sources and the downstream model's predictions should be able to distinguish the adversarial sources. In Fig. 2a we show that our system does not degrade in its initial performance, even after duplicating an adversarial LF ten times. Prior latent label models, on the other hand, rapidly degrade, given that they often assume the weak label sources to be conditionally independent given the latent label, equivalent to a Naive Bayes generative model. Note that the popular open-source implementation of [31, 32] does not support user-provided LF dependencies modeling, while [19, 11] did not converge in our experiments when modeling dependencies, and as such we were not able to test their performance when the correlation dependencies between the duplicates are provided (which in practice, of course, are not known).

We also run a synthetic experiment inspired by [9], where one LF is set to the true labels of the ProfTeacher dataset, i.e. $\lambda_1 = y^*$, while the other LF simply votes according to a coin flip, i.e. $\lambda_2 \sim P(y)$, and we then duplicate this latter LF, i.e. $\lambda_3 = \cdots = \lambda_m = \lambda_2$. Under this setting, our `WeaSEL` model is able to consistently *recover the fully supervised performance* of the same downstream model directly trained on the true labels $y^*$, *even when we duplicate the random LF up to* 2000 *times* ($m = 2001$). Snorkel and triplet methods, on the other hand, were unable to recover the true label (AUC $\approx 0.5$). Importantly, we find that the design choices for `WeaSEL` are to a large

Table 3: Dataset details, where training, validation and test set sizes are $N_{train}$, $N_{val}$, $N_{test}$ respectively, and $f$ denotes the downstream model type. We also report the total coverage Cov. of all LFs, which refers to the percentage of training samples which are labeled by at least one LF (the rest is not used). For IMDB we used two different sets of labeling functions of sizes 12 and 136.

| Dataset | #LFs | $N_{train}$ | Cov. (in %) | $N_{val}$ | $N_{test}$ | $f$ |
|---------|------|-------------|-------------|-----------|------------|-----|
| Spouse | 9 | $22,254$ | 25.8 | 2811 | 2701 | LSTM |
| BiasBios | 99 | $12,294$ | 81.8 | 250 | $12,044$ | MLP |
| IMDB | 12 | $25k$ | 88.0 | 250 | $24,750$ | MLP |
| IMDB | 136 | $25k$ | 83.1 | 250 | $24,750$ | MLP |
| Amazon | 175 | $160k$ | 65.5 | 500 | $39,500$ | MLP |

extent key in order to recover the true labels in a stable manner as in Fig. 2b. Various other choices either collapse similarly to the baselines, are not able to fully recover the supervised performance, or lead to unstable test performance curves, see Fig. 5 in the appendix. More details on the experimental design and an extensive discussion, ablation, and figures based on the synthetic experiment can be found in the appendix F.

## 4.4 Implementation Details

Here we provide a high-level overview over the used encoder architecture, the LF sets, and the features. More details, especially hyperparameter and architecture details, are provided in Appendix C. All downstream models are trained with the (binary) cross-entropy loss, and our model with the symmetric version of it that uses `stop-grad` on the targets.

**Encoder network**  The encoder network $e$ does not need to follow a specific neural network architecture and we therefore use a simple multi-layer perceptron (MLP) in our benchmark experiments.

**Features for the encoder**  A big advantage of our model is that it is able to take into account the features $\mathbf{x}$ for generating the sample-by-sample source accuracies. For all datasets, we concatenate the LF outputs with the same features that are used by the downstream model as input of our encoder model (for Spouse we use smaller embeddings than the ones used by the downstream LSTM).

**Weak supervision sources**  For the Spouse dataset, and the IMDB variant with 12 LFs, we use the same LFs as in [19, 11] respectively. The remaining three LF sets were selected by us prior to running experiments. These LFs are all pattern- and regex-based heuristics, while the Spouse experiments also contain LFs that are distant supervision sources based on DBPedia.

## 5 Ablations

In this section we demonstrate the strength of the `WeaSEL` model design decisions. We perform extensive ablations on all four main datasets but Spouse for twenty configurations of `WeaSEL` with different encoder architectures, hyperparameters, and loss functions. The tabular results and a more detailed discussion than in the following can be found in Appendix D.

We observe that ignoring the features when modeling the sample-dependent accuracies, i.e. $\theta(\boldsymbol{\lambda}, \mathbf{x}) = \theta(\boldsymbol{\lambda})$, usually underperforms by up to 1.2 F1 points. A more drastic drop in performance, up to 4.9 points, occurs when the encoder network is linear, i.e. without hidden layers, as in [9]. It also proves helpful to scale the softmax in Eq. 3 by $\sqrt{m}$ via the inverse temperature parameter $\tau_2$. Further, while the MIG loss proved important for `WeaSEL` to achieve state-of-the-art performance on the crowdsourcing dataset (with a similar lift in performance observable for Snorkel using MIG for downstream model training), this does not hold for the main datasets. This indicates that the MIG loss is a good choice for crowdsourcing, but not for more general WS settings.

Our ablations also show that it is important to restrict the accuracies to a positive interval (e.g. (0, 1), with the sigmoid function being a good alternative to the softmax we use). On the one hand, this encodes the inductive bias that LFs are not adversarial, i.e. can not have negative accuracies, (using tanh to output accuracy scores does not perform well), and on the other hand does not give the encoder

network too much freedom in the scale of the scores (using ReLU underperforms significantly as well).

Additionally, we find that our choice of using the symmetric cross-entropy loss with `stop-grad` applied to the targets is crucial for the obtained strong performance of `WeaSEL`. Removing the `stop-grad` operation, or using the standard cross-entropy (without `stop-grad` on the target) leads to significantly worse scores and a very brittle model. Losses that already are symmetric (e.g. L1 or Squared Hellinger loss) neither need to be symmetrized nor use `stop-grad`. While the L1 loss consistently underperforms, we find that the Squared Hellinger loss can lead to better performance on two of the four datasets.

However, only the symmetric cross-entropy loss with `stop-grad` on the targets is shown to be robust and able to recover the true labels in our synthetic experiment in Section 4.3. Thus, to complement the above ablation on real datasets, we additionally run extensive ablations on this synthetic setup in Appendix F. This synthetic ablation gives interesting insights, and strongly supports the proposed design of `WeaSEL`. Indeed, many choices for `WeaSEL` that perform well enough on the real datasets, such as no features for the encoder, $\tau_2 = 1$, sigmoid parameterized accuracies, and all other losses that we evaluated, lead to significantly worse performance and less robust learning on the synthetic adversarial setups.

## 6   Practical Aspects and Limitations

**On why it works & degenerate solutions**    Overall, `WeaSEL` avoids trivial overfitting and degenerate solutions by hard-coding the encoder generated labels as a (normalized) linear combination of the $m$ LF outputs, weighted by $m$ sample-dependent accuracy scores. This design choice also ensures that the randomly initialized $e$ will lead the downstream model $f$ that is trained on soft labels generated by the random encoder, to obtain performance similar to when $f$ is trained on majority vote labels. In fact, the random-encoder-`WeaSEL` variant itself often outperforms other baselines, and triplet methods in particular (see appendix B).

Empirically, we only observed degenerate solutions when training for too many epochs. Early-stopping on a small validation set ensures that a strong final solution is returned, and should be done whenever such a set exists or is easy to create. When no validation set is available, we find that choosing the temperature hyperparameter in Eq. 3 such that $\tau_1 \leq 1/3$ avoids collapsed solutions on all our datasets. This can be explained by the fact that a lower inverse temperature forces the encoder-predicted label to always depend on multiple LF votes when available, rather than a single one (which happens when the $\mathrm{softmax}$ in Eq. 3 becomes a $\mathtt{max}$ as $\tau_1 \to \infty$). This makes it harder for the encoder to overfit to individual LFs. Our ablations indicate that this temperature parameter setting comes at a small cost in terms of loss in downstream performance, compared to when using a validation set for early stopping. Thus, when no validation set is available, we advise to lower $\tau_1$.

**Complex downstream models**    We have shown that `WeaSEL` achieves competitive or state-of-the-art performance on all datasets we tried it on, for a given set of LFs. In practice, however, this LF set needs to first be defined by users. This can be done via an iterative process, where the feedback is sourced from the quality of the probabilistic labels generated by the label model. A limitation of our model, is that each such iteration would require training the downstream model, $f$. When $f$ is slow to train, this may slow down the LF development cycle and lead to unnecessary energy consumption. A practical solution to this can be to a) do the iteration cycle with a less complex downstream model; or b) use the fast to train PGM-based label models to choose a good LF set, and then move to `WeaSEL` in order to achieve better downstream performance.

## 7   Extensions

**Probabilistic labeling functions**    Our learning method can easily support labeling functions that output continuous scores instead of discrete labels as in [10]. In particular, this includes probabilistic sources that output a distribution over the potential class labels. This can be encoded in our model by changing the one-hot representation of our base model to a continuous representation $\bar{\boldsymbol{\lambda}} \in [0, 1]^{m \times C}$.

**Modeling more structure**    While we use a simple multi-layer perceptron (MLP) as our encoder $e$ in our benchmark experiments, our formulation is flexible to support arbitrarily complex networks. In particular, we can naturally model dependencies amongst weak sources via edges in a Graph Neural Network (GNN), where each LF is represented by a node that is given the LF outputs as features. Furthermore, while we only explicitly reparameterized the accuracy parameters of the sources in our base model, it is straightforward to augment $\bar{\lambda}$ with additional sufficient statistics, e.g. the fixing or priority dependencies from [30, 8] that encode that one source fixes (i.e. should be given priority over) the other whenever both vote.

## 8    Conclusion

We proposed `WeaSEL`, a new approach for end-to-end learning of neural network models for classification from, exclusively, multiple sources of weak supervision that streamlines prior latent variable models. We evaluated the proposed approach on benchmark datasets and observe that the downstream models outperform state-of-the-art data programming approaches in 4 out of 5 cases while remaining highly competitive on the remaining task, and outperforming several state-of-the-art crowdsourcing methods on a crowdsourcing task. We also demonstrated that our integrated approach can be more robust to dependencies between the labeling functions as well as to adversarial labeling scenarios. The proposed method works with discrete and probabilistic labeling functions and can utilize various neural network designs for probabilistic label generation. This end-to-end approach can simplify the process of developing effective machine learning models using weak supervision as the primary source of training signal, and help adoption of this form of learning in a wide range of practical applications.

## Acknowledgements

This work was made possible thanks to Carnegie Mellon University's Robotics Institute Summer Scholars program and was partially supported by a Space Technology Research Institutes grant from NASA's Space Technology Research Grants Program, and by Defense Advanced Research Projects Agency's awards FA8750-17-2-0130 and HR0011-21-9-0075.

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
