# Appendix

## A  Posterior Reparameterization

In this section we motivate the design choices and inductive biases that we encode into our neural encoder network $e$, which is the network that is used to model the relative accuracies of the weak supervision sources $\boldsymbol{\lambda}$. Recall that we model the probability of a particular sample $\mathbf{x} \in \mathcal{X}$ having the class label $y \in \mathcal{Y} = \{1, \ldots, C\}$ as

$$P_\theta(y \mid \boldsymbol{\lambda}) = \mathrm{softmax}\,(\mathbf{s})_y \, P(y), \tag{4}$$

$$\mathbf{s} = \theta(\boldsymbol{\lambda}, \mathbf{x})^T \bar{\boldsymbol{\lambda}} \in \mathbb{R}^C. \tag{5}$$

where $\theta(\boldsymbol{\lambda}, \mathbf{x}) \in \mathbb{R}^m$ weighs the LF votes on a sample-by-sample basis and the softmax for class $y$ on $s$ is defined as

$$\mathrm{softmax}\,(\mathbf{s})_y = \frac{\exp\left(\theta(\boldsymbol{\lambda}, \mathbf{x})^T \mathbb{1}\{\boldsymbol{\lambda} = y\}\right)}{\sum_{y' \in \mathcal{Y}} \exp\left(\theta(\boldsymbol{\lambda}, \mathbf{x})^T \mathbb{1}\{\boldsymbol{\lambda} = y'\}\right)}.$$

**Connection to prior PGM models**   We now motivate this choice by deriving a less expressive variant of it from the standard Markov Random Field (MRF) used in the related work. If we view the attention scores $\theta(\boldsymbol{\lambda}, \mathbf{x}) \in \mathbb{R}^m$, that assign sample-dependent accuracies to each labeling function, as sample-independent parameters $\theta_1$ and, by that, drop the features from the equation – as is done in the related work [30, 32, 19, 11] – we can rewrite Eq. 4 as

$$\frac{\exp\left(\theta_1^T \mathbb{1}\{\boldsymbol{\lambda} = y\}\right)}{\sum_{y' \in \mathcal{Y}} \exp\left(\theta_1^T \mathbb{1}\{\boldsymbol{\lambda} = y'\}\right)} P(y)$$

Let $\phi_1(\boldsymbol{\lambda}, y) = \mathbb{1}\{\boldsymbol{\lambda} = y\}$, and, for clarity of writing, we drop the class balance, then this becomes

$$= \frac{\exp\left(\theta_1^T \phi_1(\boldsymbol{\lambda}, y)\right)}{\sum_{y' \in \mathcal{Y}} \exp\left(\theta_1^T \phi_1(\boldsymbol{\lambda}, y')\right)}$$

$$= \frac{Z_\theta^{-1} \exp\left(\theta_1^T \phi_1(\boldsymbol{\lambda}, y) + \theta_2^T \phi_2(\boldsymbol{\lambda})\right)}{\sum_{y' \in \mathcal{Y}} Z_\theta^{-1} \exp\left(\theta_1^T \phi_1(\boldsymbol{\lambda}, y') + \theta_2^T \phi_2(\boldsymbol{\lambda})\right)}$$

$$= \frac{P_\theta(\boldsymbol{\lambda}, y)}{\sum_{y' \in \mathcal{Y}} P_\theta(\boldsymbol{\lambda}, y')}$$

$$= \frac{P_\theta(\boldsymbol{\lambda}, y)}{P_\theta(\boldsymbol{\lambda})}$$

$$= P_\theta(y \mid \boldsymbol{\lambda}),$$

where in the second step we multiplied the denominator and numerator with the same quantity $\frac{1}{Z_\theta} \exp\left(\theta_2^T \phi_2(\boldsymbol{\lambda})\right)$, and $\theta$ now parameterizes the joint distribution of the latent label and weak sources as

$$P_\theta(\boldsymbol{\lambda}, y) = \frac{1}{Z_\theta} \exp\left(\theta_1^T \phi_1(\boldsymbol{\lambda}, y) + \theta_2^T \phi_2(\boldsymbol{\lambda})\right) = \frac{1}{Z_\theta} \exp\left(\theta^T \phi(\boldsymbol{\lambda}, y)\right).$$

We can recognize $P_\theta$ as a distribution from the exponential familiy, and more specifically as a pairwise MRF, or factor graph, with canonical parameters $\theta = (\theta_1, \theta_2)$ and corresponding sufficient statistics, or factors, $\phi(\boldsymbol{\lambda}, y) = (\phi_1(\boldsymbol{\lambda}, y), \phi_2(\boldsymbol{\lambda}))$, as well as the log partition function $Z_\theta$. The accuracy factors and parameters $\phi_1, \theta_1$ are the core component of this model and sometimes take the form $\phi_1(\boldsymbol{\lambda}\, y) = \boldsymbol{\lambda}\, y$ in binary models as in [30, 19, 11]. The label-independent factors $\phi_2(\boldsymbol{\lambda})$ have, as can be seen from the derivation above, no direct influence on the latent label posterior, but are often used to model labeling propensities $\mathbb{1}\{\boldsymbol{\lambda} \neq 0\}$ and correlation dependencies $\mathbb{1}\{\lambda_i = \lambda_j\}$, which can be important for PGM parameter learning, but are susceptible to misspecifications [39, 11, 8]. *Our own parameterization therefore is a more expressive variant of these latent-variable PGM models, where we are able to assign LF accuracies on a sample-by-sample basis. Furthermore, our neural encoder network outputs them as a function of the LF outputs **and** features, and is expected to learn the easy to misspecify dependencies and label-independent statistics implicitly. Indeed, our empirical findings and subsection 4.3 support this.*

Table 4: The final test F1 performance of various multi-source weak supervision methods over seven runs, using different random seeds, are averaged out $\pm$ standard deviation. The top 2 performance scores are highlighted as **First**, **Second**. Triplet-median [11] is not listed as it only converged for IMDB with 12 LFs (F1 = $73.0 \pm 0.22$), and Spouse (F1 = $48.7 \pm 1.0$). Sup. (Val. set) is the performance of the downstream model trained in a supervised manner on the labeled validation set. The rest are state-of-the-art latent label models. For reference, we also report the *Ground truth* performance of a fully supervised model trained on true training labels (which are unused by all other models, and not available for Spouse). We also report the performance of `WeaSEL-random`, where only the downstream model of `WeaSEL` is trained (and the encoder network is left at its randomly initialized state). All models are run twice, where only the learning rate differs (either $10^{-4}$ or $4 \cdot 10^{-5}$), and the model with best ROC-AUC on the validation set is reported. The probabilistic labels from Snorkel used for downstream model training are chosen over six different configurations of the learning rate and number of epochs (again with respect to validation set ROC-AUC).

| Model | Spouse (9 LFs) | ProfTeacher (99 LFs) | IMDB (136 LFs) | IMDB (12 LFs) | Amazon (175 LFs) |
|---|---|---|---|---|---|
| Ground truth | – | $90.65 \pm 0.29$ | $86.72 \pm 0.40$ | $86.72 \pm 0.40$ | $92.93 \pm 0.68$ |
| Sup. (Val. set) | $20.4 \pm 0.2$ | $73.34 \pm 0.00$ | $68.76 \pm 0.00$ | $68.76 \pm 0.00$ | $84.18 \pm 0.00$ |
| Snorkel | $48.79 \pm 2.69$ | $85.12 \pm 0.54$ | $\mathbf{82.22 \pm 0.18}$ | $\mathbf{74.45 \pm 0.58}$ | $80.54 \pm 0.41$ |
| Triplet | $45.88 \pm 3.64$ | $74.43 \pm 10.59$ | $75.36 \pm 1.92$ | $73.15 \pm 0.95$ | $75.44 \pm 3.21$ |
| Triplet-Mean | $\mathbf{49.94 \pm 1.47}$ | $82.58 \pm 0.32$ | $79.03 \pm 0.26$ | $73.18 \pm 0.23$ | $79.44 \pm 0.68$ |
| WeaSEL-random | $46.43 \pm 3.29$ | $83.47 \pm 0.64$ | $79.80 \pm 0.48$ | $74.22 \pm 0.45$ | $82.22 \pm 0.57$ |
| Majority vote | $40.67 \pm 2.01$ | $\mathbf{85.44 \pm 0.37}$ | $80.86 \pm 0.28$ | $74.13 \pm 0.31$ | $\mathbf{84.20 \pm 0.52}$ |
| WeaSEL | $\mathbf{51.98 \pm 1.60}$ | $\mathbf{86.98 \pm 0.45}$ | $\mathbf{82.10 \pm 0.45}$ | $\mathbf{77.22 \pm 1.02}$ | $\mathbf{86.60 \pm 0.71}$ |

# B  Extended Results

We provide more detailed results in Table 4. Here, we include `WeaSEL-random`, which corresponds to `WeaSEL` with a randomly initialized encoder network that is not trained/updated. As expected, this setting produces performance often similar compared to training an end model on the hard majority vote labels. This is due to the strong inductive bias in our encoder model that constrains the encoder labels to be a normalized linear combination of the LF votes, weighted by positive accuracy scores. In fact, `WeaSEL-random` itself is often able to outperform the PGM-based baselines, in particular the triplet methods. Our results show that `WeaSEL` consistently improves significantly upon these baselines via training the encoder network to maximize its agreement with the downstream model.

# C  Extended Implementation Details

**Weak supervision sources**    For the Spouses dataset, and the IMDB variant with 12 LFs, we use the same LFs as in [19] and [11], respectively[5]. The set of 12 IMDB LFs was specifically chosen to have a large coverage, see Table 3. These LFs and the larger set of LFs that we introduce for the second IMDB experiment are all pattern- and regex-based heuristics, i.e. LFs that label whenever a certain word or bi-gram appears in a text document. For instance, 'excellent' would label for the positive movie review sentiment (and would do so with $80\%$ accuracy on the samples where it does not abstain). This holds for the other text datasets as well, while the Spouse experiments also contain LFs that are distant supervision sources based on DBPedia.
For the remaining datasets (IMDB with 136 LFs, Bias Bios, and Amazon), we created the respective LF sets ourselves, prior to running experiments.

**Encoder network architectures**    In all experiments, we use a simple multi-layer perceptron (MLP) as the encoder $e$, with two hidden layers, batch normalization, and ReLU activation functions. For the Spouse dataset, we use a bottleneck-structured network of sizes 50, 5. This is motivated by the small size of the set of samples labeled by at least one LF. For all other datasets we use hidden dimensions of 70, 70. We show in the ablations (Table 5), that our end-to-end model also succeeds for different encoder architecture choices.

---

[5]All necessary label matrices are available in our research source code. The Spouse LFs and data are also available at the following URL: `https://github.com/snorkel-team/snorkel-tutorials/blob/master/spouse/spouse_demo.ipynb`

**Downstream models** For all datasets besides Spouse, we use a three-layer MLP with hidden dimensions 50, 50, 25. For Spouse, we use a single-layer bidirectional LSTM with a hidden dimension of 150, followed by two fully-connected readout layers with dimensions 64, 32. All fully-connected, layers use ReLU activation functions. We choose simple downstream architectures as we are interested in the relative improvements over other label models. More sophisticated architectures are expected to further improve the performances, however.

**Hyperparameters** Unless explicitly mentioned, all reported experiments are averaged out over seven random seeds. We use an L2 weight decay of 7e-7 and dropout of $0.3$ for both encoder and downstream model for all datasets but Spouse (where the LSTM does not use dropout). All models are optimized with Adam, with early-stopping based on AUC performance on the small validation set, and a maximum number of $150$ epochs ($75$ for Spouse). The batch size is set to $64$. The loss function is set to the (binary) cross-entropy. For each dataset and each model/baseline, we run the same experiment for learning rates of 1e-4 and 3e-5, and then report the model chosen according to the best ROC-AUC performance on the small validation set. For Spouse we additionally run experiments with a L2 weight decay of $1$e-4 which due to the risk of overfitting to the small size of LF-covered data points boosts performance for all models. For our own model, WeaSEL, we also run additional experiments for Spouses with different configurations of the temperature hyperparameter, $\tau_1 \in \{1, 1/3\}$ and again report the test performance as measured by the best validation ROC-AUC. The probabilistic labels from Snorkel used for downstream model training are chosen over six different configurations of the learning rate and number of epochs for Snorkel's label model (again with respect to validation set ROC-AUC). For all binary classification datasets (i.e. all except for LabelMe), we tune the downstream model's decision threshold based on the resulting F1 validation score for all models. We believe that this, alternatively to reporting test ROC-AUC scores, makes the comparison fairer, since F1 is a threshold dependent metric. All label model baselines are provided with the class balance, which WeaSEL does not use (but which is expected to be helpful for unbalanced classes, where no validation set is available).

# D  Extended Ablations

The full ablations are reported in Table 5, where in each row we change or remove exactly one component of our proposed model, WeaSEL. We find that the design choices of WeaSEL which were inspired by sensible inductive biases for an encoder label model are hard to beat by various changes to the architecture, loss function, or hyperparameters. Indeed, most changes consistently underperform WeaSEL, and the occasional positive changes – 1e-4 weight decay, and the Squared Hellinger loss instead of the symmetric cross-entropy – only beat the base WeaSEL performance in at most two datasets, and never significantly. In practice, we advise to explore these strongest configurations if a small validation set is available.

We find that letting the accuracy scores depend on the input features (first row), usually boosts performance, but not by much (1.2 F1 points at most). On the other hand, it proves very important to allow these accuracy scores to depend non-linearly on the LF votes and the features: A linear encoder network, as in [9], significantly underperforms WeaSEL with at least one hidden layer by up to 4.9 F1 score points. Conversely, a deeper encoder network (of hidden dimensionalities $75, 50, 25, 50, 75$, see fourth row) does not improve results. This may be due to the sample-dependent accuracies not being a too complex function to learn.

While the effect of the inverse temperature parameter $\tau_1$–which controls the softness of the encoder-predicted accuracy scores–on downstream performance is not large, it can have significant effects on the learning dynamics and robustness, see Fig 3 for such learning curves as a function of epoch number. In particular, a lower $\tau_1$ makes the dynamics more robust, since the accuracy score weights are more evenly distributed across LFs, which appears to help avoid overfitting. When overfitting is not easily detectable due to a lack of a validation set, it is therefore advisable to use a lower $\tau_1$. It also proves helpful to scale the softmax in Eq. 3 by $\sqrt{m}$, rather than not scaling it ($\tau_2 = 1$ row) or scaling by $m$.

Changing the loss function from the symmetric cross-entropy to the MIG function [9] or the L1 loss consistently leads to worse performance. The former is interesting, since using the MIG loss for the crowdsourcing dataset LabelMe, see subsection 4.2, was important in order to achieve state-of-the-art crowdsourcing performance (with a similar lift in performance observable for Snorkel using MIG for downstream model training). The result provides some evidence that the MIG loss

Table 5: Ablative study on the subcomponents of our algorithm as in Alg. 1 (over 5 random seeds). In each row below we change exactly one component of `WeaSEL` and report the resulting F1 score. Note that the scores for `WeaSEL` are slightly different to the ones in the main results table, since they were run separately, with fewer seeds, and for only one learning rate (1e-4). Configurations that **outperform base `WeaSEL` are highlighted in bold font**, while the **four worst performing configurations** are highlighted in red for each dataset. Note that bold font does not indicate significant differences.

| Change | ProfTeacher | IMDB-136 LFs | IMDB-12 LFs | Amazon |
|---|---|---|---|---|
| `WeaSEL` | $86.8 \pm 0.4$ | $82.1 \pm 0.7$ | $77.3 \pm 0.5$ | $86.6 \pm 0.5$ |
| $\theta(\boldsymbol{\lambda}, \mathbf{x}) = \theta(\boldsymbol{\lambda})$ | $85.6 \pm 1.6$ | $82.1 \pm 0.5$ | $75.9 \pm 0.8$ | $86.6 \pm 0.4$ |
| Linear $e$ | $81.9 \pm 0.7$ | $80.0 \pm 0.6$ | $73.2 \pm 0.6$ | $82.6 \pm 0.5$ |
| 1 hidden layer $e$ | $\mathbf{87.1 \pm 0.7}$ | $81.8 \pm 0.6$ | $76.8 \pm 0.9$ | $85.3 \pm 0.8$ |
| 75x50x25x50x75 $e$ | $84.3 \pm 2.1$ | $81.9 \pm 0.6$ | $75.8 \pm 1.1$ | $86.1 \pm 0.6$ |
| $\tau_1 = 2$ | $86.7 \pm 1.0$ | $81.9 \pm 0.3$ | $77.3 \pm 0.5$ | $85.5 \pm 1.0$ |
| $\tau_1 = 1/2$ | $86.5 \pm 0.8$ | $81.8 \pm 0.5$ | $76.0 \pm 1.4$ | $86.4 \pm 0.3$ |
| $\tau_1 = 1/4$ | $84.5 \pm 1.2$ | $81.8 \pm 0.2$ | $73.9 \pm 0.9$ | $85.6 \pm 1.0$ |
| $\tau_2 = 1$ | $85.2 \pm 1.6$ | $\mathbf{82.2 \pm 0.4}$ | $76.6 \pm 1.0$ | $84.3 \pm 1.2$ |
| $\tau_2 = m$ | $86.1 \pm 0.7$ | $81.2 \pm 0.6$ | $76.4 \pm 0.4$ | $85.7 \pm 0.2$ |
| No BatchNorm | $82.6 \pm 1.4$ | $81.9 \pm 0.5$ | $74.7 \pm 0.7$ | $85.3 \pm 0.8$ |
| 1e-4 weight decay | $\mathbf{87.4 \pm 0.4}$ | $80.9 \pm 1.3$ | $\mathbf{77.9 \pm 0.6}$ | $85.2 \pm 0.5$ |
| MIG loss | $86.7 \pm 0.4$ | $78.7 \pm 0.4$ | $74.1 \pm 0.4$ | $84.7 \pm 1.8$ |
| L1 loss | $86.2 \pm 0.6$ | $81.1 \pm 0.5$ | $75.6 \pm 0.9$ | $84.1 \pm 0.9$ |
| Squared Hellinger loss | $\mathbf{87.4 \pm 0.3}$ | $\mathbf{82.2 \pm 0.6}$ | $75.7 \pm 1.1$ | $86.3 \pm 0.4$ |
| $CE(P_f, P_e)$ asymm. loss | $\color{red}\mathbf{77.3 \pm 3.7}$ | $\color{red}\mathbf{77.7 \pm 1.1}$ | $71.7 \pm 0.3$ | $\color{red}\mathbf{78.7 \pm 1.2}$ |
| $CE(P_e, P_f)$ asymm. loss | $\color{red}\mathbf{73.1 \pm 6.8}$ | $\color{red}\mathbf{71.9 \pm 1.9}$ | $\color{red}\mathbf{69.7 \pm 0.7}$ | $\color{red}\mathbf{70.1 \pm 1.1}$ |
| No `stop-grad` | $\color{red}\mathbf{80.4 \pm 2.1}$ | $\color{red}\mathbf{76.2 \pm 0.5}$ | $\color{red}\mathbf{71.0 \pm 0.6}$ | $79.3 \pm 0.6$ |
| $\theta(\boldsymbol{\lambda}, \mathbf{x}) = \sqrt{m} \cdot \text{sigmoid}(e(\boldsymbol{\lambda}, \mathbf{x}))$ | $85.5 \pm 0.6$ | $81.8 \pm 0.5$ | $\mathbf{78.0 \pm 0.7}$ | $\mathbf{86.9 \pm 0.3}$ |
| $\theta(\boldsymbol{\lambda}, \mathbf{x}) = \text{ReLU}(e(\boldsymbol{\lambda}, \mathbf{x})) + 1\text{e-}5$ | $83.0 \pm 2.3$ | $78.3 \pm 1.1$ | $\color{red}\mathbf{69.1 \pm 2.1}$ | $\color{red}\mathbf{74.2 \pm 2.7}$ |
| $\theta(\boldsymbol{\lambda}, \mathbf{x}) = \text{Tanh}(e(\boldsymbol{\lambda}, \mathbf{x}))$ | $\color{red}\mathbf{71.9 \pm 4.0}$ | $\color{red}\mathbf{67.0 \pm 0.8}$ | $\color{red}\mathbf{67.0 \pm 1.1}$ | $\color{red}\mathbf{67.3 \pm 1.1}$ |

may be inappropiate for weak supervision settings other than crowdsourcing, while its use may be recommended for that specific setting.

We find that it is important to constrain the accuracy score space to a positive interval, either by viewing them as an aggregation of the LFs via the scaled $\text{softmax}$ in Eq. 3, or by replacing the $\text{softmax}$ with a sigmoid function. Indeed, using a less constrained activation function for the estimated accuracies (last two rows, where the 1e-5 in the ReLU row avoids accuracy scores equal to zero) significantly underperforms: Allowing the accuracies to be negative (last row) leads to collapse and bad downstream performance. This is likely due to the removal of the inductive bias that LFs are better-than-random, which makes the joint optimization more likely to find trivial solutions. Additionally, we find that our choice of using the symmetric cross-entropy loss with `stop-grad` applied to the targets is crucial for the strong performance of `WeaSEL`. Removing the `stop-grad` operation, or using the standard cross-entropy (without `stop-grad` on the target) leads to significantly worse scores and a very brittle model. This is somewhat expected, since conceptually our goal is to have an objective that maximizes the agreement between a pair of models that predict based on two different views of the latent label, the features and the LF votes. The cross-entropy with `stop-grad` on the target[6] naturally encodes this understanding, since each model uses the other model's predictions as a reference distribution. Losses that already are symmetric (e.g. L1 or Squared Hellinger loss) neither need to be symmetrized nor use `stop-grad`. While the L1 loss consistently underperforms, we find that the Squared Hellinger loss can lead to better performance on two out of four datasets.

However, only the symmetric cross-entropy loss with `stop-grad` on the targets is shown to be robust and able to recover the true labels in our synthetic experiments in appendix F, see Fig. 5 in particular. The synthetic ablation in appendix F gives interesting insights, and strongly supports the proposed design of `WeaSEL`. Indeed, many choices for `WeaSEL` that perform well enough on the real datasets,

---

[6]or, due to the `stop-grad` operation, equivalently the KL divergence

such as no features for the encoder, $\tau_2 = 1$, sigmoid parameterized accuracies, and all other objectives that we evaluated, lead to significantly worse performance and less robust learning on the synthetic adversarial setups.

## E   Crowdsourcing dataset

As the crowdsourcing dataset, we choose the multi-class LabelMe image classification dataset that was previously used in the most related crowdsourcing literature [35, 9]. Note that this dataset consists of $10k$ samples, of which only $1k$ are unique, in the sense that the rest are augmented versions of the $1k$. They were annotated by $59$ crowdworkers, with a mean overlap of $2.55$ annotations per image. The downstream model is identical to the previously reported one [35, 9]. That is, a VGG-16 neural network is used as feature extractor, and a single fully-connected layer (with 128 units and ReLU activation) and one output layer is put on top, using $50\%$ dropout.

Experiments were conducted over seven random seeds with a learning rate of 1e-4 and 50 epochs. The reported scores are the ones with best validation set accuracy for a L2 weight decay $\in \{$ 7e-7, 1e-4 $\}$. The validation set is of size 200, and was split at random from the training set prior to running the experiments.

As is usual in the related work for multi-class settings [31], we employ class-conditional accuracies $\theta(\boldsymbol{\lambda}, \mathbf{x}) \in \mathbb{R}^{m \times C}$ instead of only $m$ class-independent accuracies. Recall the LF outputs indicator matrix, $\bar{\boldsymbol{\lambda}} \in \mathbb{R}^{m \times C}$. To compute the resulting output softmax logits $\mathbf{s} \in \mathbb{R}^C$, we set $\mathbf{A} = \theta(\boldsymbol{\lambda}, \mathbf{x}) \odot \bar{\boldsymbol{\lambda}} \in \mathbb{R}^{m \times C}$ and $\mathbf{s}_j = \sum_i \mathbf{A}_{ij} \in \mathbb{R}$, where $\odot$ is the element-wise matrix product and we sum up the resulting matrix $\mathbf{A}$ across the LF votes dimension.

Snorkel+MIG indicates that the downstream model $f$ was trained on the MIG loss with respect to soft labels generated by the first Snorkel step, label modeling. Snorkel+CE refers analogously to the same training setup, but using the cross-entropy (CE) loss. All crowdsourcing baseline models are based on the open-source code from [9].

## F   Robustness experiments

In this section we give more details on the experiments that validate the robustness of our approach against (strongly) correlated LFs that are not better than a random coin flip. In addition, we present one further experiment where the random LFs are independent of each other – a more difficult setup for learning (but which does not violate any assumptions of the PGM-based methods) – and our model, WeaSEL, again is shown to be robust to a large extent.

In contrast to WeaSEL, prior PGM-based work [31, 19, 11] attain significantly worse performance under these settings, due to assuming a Naive Bayes generative model where the weak label sources are conditionally independent given the latent label.

### F.1   Adversarial LF duplication

For this experiment we use our set of 12 LFs for the IMDB dataset and generate a fake adversarial source by flipping the abstain votes, of the 80%-accurate LF that labels for the positive sentiment on 'excellent', to negative ones.

### F.2   Recovery of true labels under massive LF noise

In this set of synthetic experiments we again validate the robustness of our approach. We focus on the Bias in Bios dataset, and use the features and true labels, $y^*$, therein. We let our initial LF set consist of 1) a 100% accurate LF, that is we set $\lambda_1 = y^*$, and 2) a LF that votes according to the class balance (i.e. a coin flip with probabilities for tail/head set according to the class balance), i.e. $\lambda_2 \sim P(y)$. In the first experiment we then add the same random LF $\lambda_2$ multiple times into the LF set (i.e. we duplicate it), see F.2.1, while in the second one, we incrementally add random LFs independently of $\lambda_2$ (and independently of any other LF already in the LF set), see F.2.2. For both setups, our model, WeaSEL, is able to recover the performance of the same downstream model, $f$, that is directly trained on the true labels, $y^*$ (F1 = $90.65$, ROC-AUC = $0.967$, see Table 4). In contrast, the PGM-based baselines quickly collapse.

**Test ROC-AUC**

— \tau_1 = 0.2    — \tau_1 = 2    — \tau_1 = 0.33    — \tau_1 = 1 (E2E)

Figure 3: Test AUC performance at each training epoch for different choices of $\tau_1 \in \{1/5, 1/3, 1, 2\}$ on our synthetic experiment, see appendix F.2.1, averaged out over the number of duplicates and five random seeds. A lower $\tau_1$ leads to slower or worse convergence in this specific case. A lower $\tau_1$ corresponds to smoother accuracies, which makes their induced label depend on more LFs. Since in this specific case only one LF is 100% accurate and the rest are not better than a coin flip, the shown behavior is expected.

### F.2.1    Random LF duplication

This experiment is inspired by the theoretical comparison in Appendix E of [9] between the authors' end-to-end system and maximum likelihood estimation (MLE) approaches that assume mutually independent LFs. The authors show that such MLE methods are not robust against the following simple example with correlated LFs. Based on the setup described above in F.2, we duplicate the random LF $\lambda_2$ multiple times, i.e. $\lambda_3 = \cdots = \lambda_m = \lambda_2$. We run experiments for varying number of duplicates $\in \{2, 25, 100, 500, 2000\}$. With this synthetic set of $m$ LFs, where one LF is 100% accurate while the other $m - 1$ LFs are just as good as a random guess, we train WeaSEL in the usual way on the features from the Bias in Bios dataset as well as the corresponding, just created, LF votes. WeaSEL is able to consistently and almost completely *recover this fully supervised performance, even when the number of duplicates is very high* ($m = 2001$). Snorkel and triplets methods, on the other hand, fare far worse (AUC $\approx 0.5$) for all numbers of duplicates. This behavior is similar to the one observed in F.1 (see Fig. 2 for the performance of the baselines and WeaSEL averaged out over the varying number of duplicates, and Fig. 5a-c for the separate performance of WeaSEL for each number of duplicates).

We also run an additional ablation study on this synthetic experiment that shows that the observed robustness does not hold for all configurations of WeaSEL. In Fig. 5 we plot the test performance curves over the training epochs for each number of LF duplications.
Our proposed model, WeaSEL enjoys a stable and robust test curve (Fig. 5c) and quickly recovers the fully supervised performance, even with 2000 LF duplicates (although convergence becomes slower as the LF set contains more duplicates). On the other hand, we find that many other configurations and designs of WeaSEL lead to less robust and worse converging curves, collapses or bad performances. Indeed, for this experiment it is key to use as the loss function the proposed symmetric cross-entropy with stop-grad applied to the targets (see Fig. 5e, 5f), accuracies parameterized by a scaled (Fig. 5h) softmax (Fig. 5g), and, to a lesser extent, using the features an input to the encoder (Fig. 5d).
While the impact of not using stop-grad, or using an asymmetric cross-entropy loss is similarly bad in the main ablations on our real datasets, other configurations, and in particular sigmoid-parameterized accuracies (the choice in [25]), an unscaled softmax, and no features for the encoder, often perform well there. This additional ablation, however, provides support for why the good performances on the real datasets notwithstanding, our proposed design choices are most appropriate in order to attain strong test performances as well as stable and robust learning.

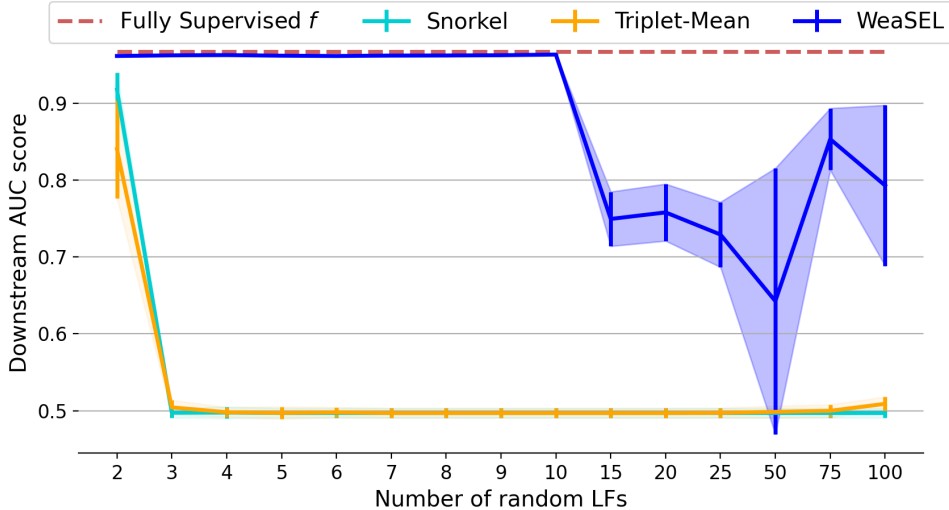

Figure 4: We start with a 100% accurate LF (i.e. ground truth labels) and incrementally add new, independent LFs that are no better than a random guess. `WeaSEL` recovers the performance of training directly on the ground truth labels (Fully Supervised $f$), for up to 10 such randomly voting LFs that are independent of each other. The PGM-based prior work, rapidly degrades in performance (AUC $\approx 0.5$) and is not able to recover any of the 100% accurate signal of the true-labels-LF, as soon as the LF set is corrupted by three or more random LFs. Performances are averaged out over five random seeds, and the standard deviation is shaded. For more details, see F.2.2

### F.2.2 Random, independent LFs

We start with the same setup as above in F.2, but instead of duplicating the same LF multiple times as in F.2.1, we now draw a new, independent random LF at each iteration. That is, we start with $\lambda_1 = y^*, \lambda_2 \sim P(y)$ as our initial LFs, and the incrementally add new LFs $\lambda_i \sim P(y)$ that have no better skill than a coin flip. Note that this is arguably a harder setup than the one in the previous experiments, since there the LF set was corrupted by a single LF voting pattern. In this experiment, multiple equally bad, but independent, LFs corrupt the 100% accurate signal of $\lambda_1$. Notably, since these $\lambda_2, \ldots, \lambda_m$ are independent, we are not violating the independence assumptions of PGM-based methods. Nonetheless, we find that these PGM-based baselines break with only three ($m = 4$) of such random, but independent LFs, while `WeaSEL` is shown to be fully robust and able to recover the ground truth LF $\lambda_1$ for up to 10 random LFs ($m = 11$). For more LFs, `WeaSEL` starts deteriorating in performance, but is still able to consistently outperform the trivial solution of voting randomly according to the class balance (i.e. based on $\lambda_2, \ldots, \lambda_m$) and the baselines, see Fig. 4.

## G Broader Impact

Large labeled datasets are important to many machine learning applications. Reducing the expensive human effort required to annotate such datasets is an important step towards making machine learning more accessible, more manageable, more beneficial, and therefore used more broadly. Our proposed end-to-end learning for weak supervision approach provides another step towards the practical utility of learning from multiple sources of weak labels on large datasets. Methods such as the one presented in our paper must be applied with care. One of the risks to consider and mitigate in a particular application is the possibility of incorporating biases from subjective humans who chose weak labeling sources. This is particularly the case when heuristics might apply differently to different subgroups in data, such as may be the case in scenarios highlighted in recent research towards fairness in machine learning.

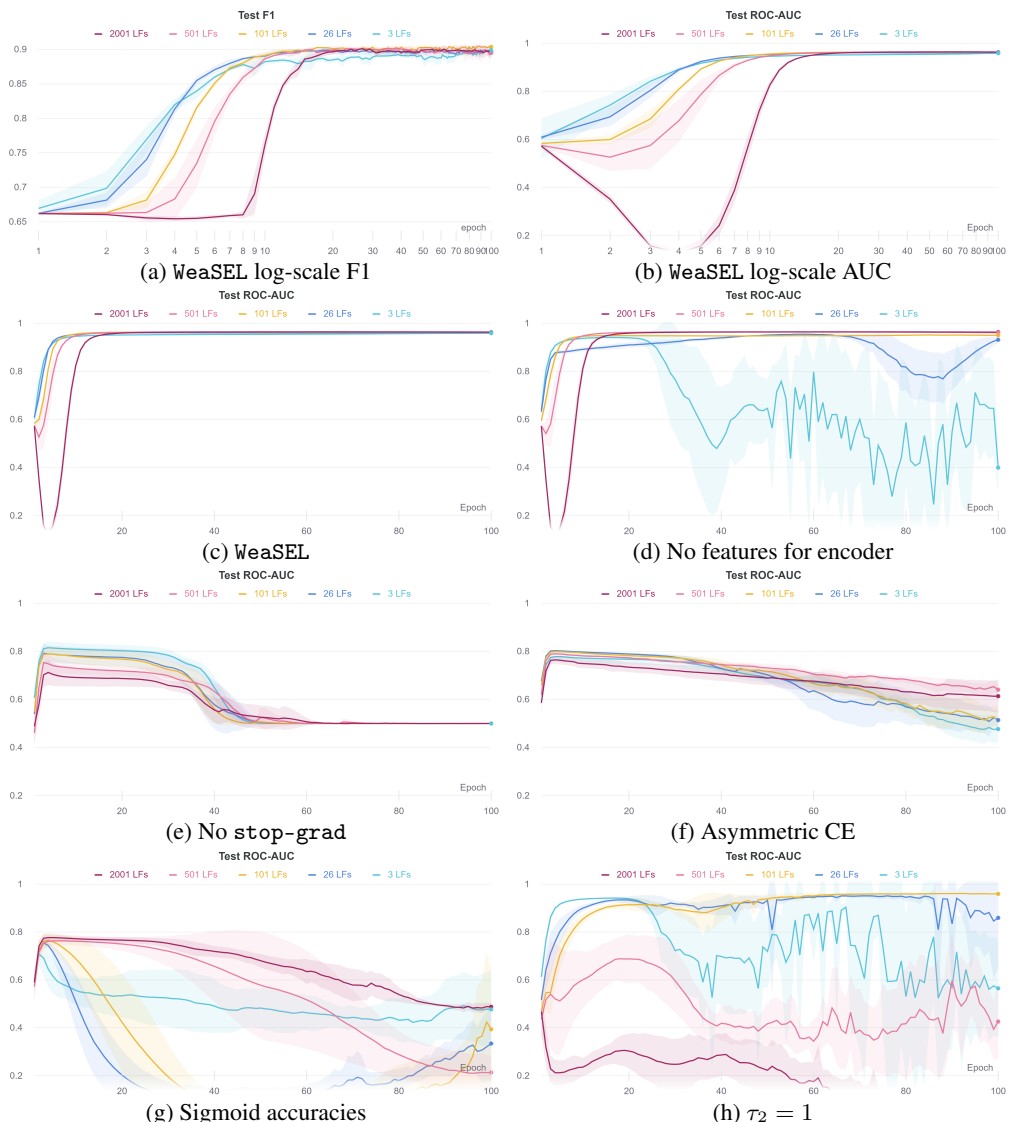

Figure 5: We start with a 100% accurate LF (i.e. ground truth labels) and plot test performances at each training epoch for a varying number of duplicates $\in \{2, 25, 100, 500, 2000\}$ of a LF that is no better than a coin flip. Performances are averaged out over five random seeds, and the standard deviation is shaded. More details are given in F.2.1.