# OpenReview forum: "End-to-End Weak Supervision"
_NeurIPS.cc/2021/Conference — NeurIPS 2021 Poster_

### Official Review · Reviewer_zED9 · 2021-07-12

**Rating:** 6
**Confidence:** 4

**Summary:**

This paper proposes a new method to training networks with multiple weak labeling sources. In contrast to previous weak supervision approaches, which aggregate weak sources using a PGM *then* train a downstream classification model, the proposed method trains an encoder network (which learns to aggregate weak sources) in tandem with a classification model (which learns to directly classify input datapoints). The authors compare their proposed method to existing weak supervision tools and find their method leads to improved performance as well as improved robustness to adversarial and correlated weak labeling sources.



**Limitations And Societal Impact:**

Both were discussed appropriately.

**Main Review:**

**Review summary.** This is an interesting and new approach to training networks with multiple weak labeling sources that seems to outperform existing weak supervision tools on the datasets tested in this submission. There are advantages of the proposed method---particularly, the ability to easily handle probabilistic weak labeling sources and correlated/adversarial sources. However, the paper lacks sufficient expository content to understand exactly why the proposed method outperforms baselines; the authors discuss a number of potential reasons, but do not back up these reasons with theoretical analyses or focused experiments. Regardless, the novelty of the method along with the improved empirical performance are strong enough to lead to my rating of marginally above the acceptance threshold.

**Positives.**
* The approach seems to handle strongly correlated and adversarial labeling functions significantly better than baseline approaches, and the structure of the correlation among sources does not need to be specified by the user a priori (unlike many existing tools, which require the user to specify the structure of the weak labeling sources). Additionally, the proposed method should be able to easily handle probabilistic weak labeling sources, which cannot be used as labeling sources in many existing weak supervision frameworks. Finally, even in non-adversarial settings with binary labeling functions, the proposed method seems to outperform existing weak supervision tools. These practical and empirical improvements to the weak supervision framework will likely benefit many users of weak supervision.
* The proposed method is, to my knowledge, novel and departs from the approach of most existing weak supervision tools. The method is sufficiently described, code is supplied, and the code will be made public (according to the authors). The method is validated with experiments on standard weak supervision/crowdsourcing benchmarks with existing state-of-the-art weak supervision/crowdsourcing tools, with synthetic datasets (although the synthetic settings are quite extreme), and via ablations.
* The paper is clear, well-written, and well-organized. Similarly, the code is commented and easy to follow.

**Limitations.**
* One of the advantages of using a PGM to model the weak labeling sources is that the PGM can be theoretically analyzed (e.g., to produce bounds on the generalization risk, understand conditions under which the model parameters can be recovered exactly, analyze how the weakly supervised performance scales with unlabeled data). These theoretical underpinnings have supported the use and understanding of weak supervision. However, the authors of this submission do not provide any theoretical support for understanding their method nor do they discuss which---if any---previous weak supervision analyses extend to their method. As a result, it’s difficult to understand exactly why the proposed method is outperforming the existing approaches.
* Even if the authors do not present theory analyzing why their proposed method works, there are many analyses they could perform to dig into some of their intuition. For example, the following experiments would be useful to help the reader understand why the proposed approach leads to improved empirical performance:
    * The authors claim that the improved empirical performance could be due to the fact that the label model parameters can adapt to each specific sample. A simple analysis here would be to look at the difference in thetas (the label model weighting parameters) for each sample in your dataset after convergence---how much are the weighting parameters actually changing? Does this correspond with relative improvement you see over baseline weak supervision methods?
    * Similarly, the authors suggest their model may be performing better because the assumptions in the standard weak supervision models don't hold. It would be interesting to run an experiment on synthetic datasets analyzing the relative improvement of the proposed method as labeling sources become increasingly correlated, breaking independence assumptions in the existing models.
    * Additional scaling experiments on the size of the weakly labeled dataset (comparing E2E, Snorkel, and Triplet) would be useful. It’s unclear to me if the proposed method will require more unlabeled data than baseline methods since more parameters need to be learned. The authors seem to be thinking along similar lines, as they hypothesize, “Since our model is based on a neural network, we hypothesize that the large relative lift in performance on the Amazon review dataset is due to it being the largest dataset size on which we evaluate on – we expect this lift to hold or become larger as the training set size increases.” However, they don’t do experiments to validate this claim. It would be relatively easy to use 10%, 20%, … 90%, 100% of the weakly labeled data with the proposed method and various baselines to generate intuition for how each method’s performance scales with more weakly labeled data.
 * As mentioned by the authors, training the label model and end model in tandem may hamper the labeling function development process. One advantage to the previous “two-phase” approach (where a graphical model is used to aggregate the labeling sources into probabilistic labels, then the weakly-labeled training data is used to train a downstream model) is its flexibility: labeling functions can be developed, iterated on, aggregated, and the resulting weak labels can be analyzed without having to wait for a deep model to train; then, the resulting labeled training set can be used with different loss functions, networks, etc. downstream. By combining the weak label aggregation process with the classification model, users are a more constrained and may not progress through this process as quickly or flexibly. However, the authors also acknowledge this as a limitation in their submission and suggest a reasonable workaround.
* Some of the motivation for the proposed approach is a little tenuous. Specifically, in the Introduction, the authors set up that existing weak supervision models suffer from the drawback that “the separate PGM does not take the predictions of the end model into account.” However, it is not clear or intuitive to me *why* the label model should take into account the predictions of the end model. Additionally, the authors focus on the fact that “current approaches for estimating the unknown class label via a PGM need to rely on computationally expensive approximate sampling methods, estimation of the full inverse of the LFs covariance matrix…” suggesting that existing tools need to be more computationally efficient. However, the proposed method is to learn an MLP in tandem with a deep neural network from large datasets, which is not particularly computationally efficient. Additionally, there are no experiments on the relative efficiency of different weak supervision methods, so this motivation in the Introduction seems unrelated to the proposed solution. In my opinion, the motivation around modelling assumptions and model misspecification is stronger and probably sufficient.


**Minor comments and clarifications**
* In Table 1, I believe you are reporting the performance of a downstream model trained on the probabilistic labels produced by Snorkel, Triplet, Triplet-mean, and Majority vote (as opposed to reporting the label model performance). However, I had to look at your code to determine this---it wasn’t clear from the writing. If you are indeed reporting the end model performance in Table 1, that should be made more clear in the writing.
* Line 195-198 was written in a confusing way---it’s not clear to me what dataset splits are used for which datasets, what data was unlabeled vs. labeled. Recommend rewording for clarity.
* It would be interesting to look at the correlations of the probabilistic labels produced by the proposed method and baseline weak supervision methods---is the proposed method catching some particular slice of the input data that the baseline methods are not, or is the proposed method just uniformly better at predicting probabilistic labels?
* It would be helpful to list the performance of each labeling function and each label model (even if only on a subset of the validation or test dataset) instead of only the final classification score.
* Line 276---what is 2b? Do you mean Figure 2b?
* Formatting: there is no space between paragraphs, which is useful for the reader to guide the eye---recommend adding if possible.
* Some of the citation formatting seems to be off (e.g., [37])


----------------------------------------------------------------------------------------------------------------------------------------------------------
UPDATE AFTER REBUTTAL

I have read through all other reviews and the authors rebuttal. I appreciate the authors thorough responses to my questions and the additional experimental results---the scaling experiments with different sizes of training data were interesting to look at! Adding a bit of related work about the benefits of joint optimization as previously investigated in crowd sourcing would be useful (if it isn’t already there). As brought up by the other reviewers and by the authors themselves, there is more work to be done understanding why this approach works. In my opinion, though, I think the novel approach and strong empirical results are interesting enough for publication at NeurIPS, and I am sticking with my original review score.

**Time Spent Reviewing:**

3

---

> ### Author Response · Authors · 2021-08-11
> **Response #1 to Reviewer zED9**
>
> First of all, we would like to thank you very much for your thoughtful and detailed questions as well as feedback.
>
>
> ### PGM assumptions and synthetic correlation experiment
>
> - We did report a synthetic experiment in the manuscript, where we increased the number of correlated (identical, in the experiment) LFs. Figure 2b) shows the learning curve of E2E, baselines and a fully supervised model, averaged out over the number of LF duplications \in {2, 5, …, 2000}. As can be seen, only E2E remains robust against such extreme LF correlation.
> - Similarly, Fig. 2a) plots downstream performance against the number of correlated/duplicated worse-than-random LFs added to our IMDB-12 set. Again, we see that E2E remains robust for all numbers of duplicates (up to 10 in the plot), while Snorkel and triplet-methods break after only 3 duplicates.
>
> Do you have any suggestions for similar experiments in mind?
>
>
> ### Data Scaling experiments
> Thanks for the great suggestion! We ran such data scaling experiments on 4 of our main datasets, and plot the end-model test F1 score as a function of the number of data points used for training (written as the fraction of total available training points, over 5 random seeds with default hyperparameters and identical end-models). The plots can be found here: https://drive.google.com/drive/folders/1IiNGVk2NLumanu-zUwMuWzpIL0YGkL78?usp=sharing
>
> It is worth noting that all datasets but Amazon have a relatively small training set that makes it hard to say how predictive these plots are. However, we can observe that
> - E2E seems to scale similarly to the end-model trained on true labels in a supervised manner.
> - At least on these datasets, E2E achieves comparable performance to using the full training set with only (up to) 50% of it
> - Even in the low-data regime of just a few thousand data points, E2E outperforms Snorkel on all datasets but ProfTeacher.
>
> ### Motivation for taking the end-model's predictions into account/joint optimization
> - Taking the end-model predictions into account translates to using the additional discriminative power of the
> end-model *and* the features (which are otherwise ignored for label modeling).
> - Importantly, when our goal is to train an end-model, we do not really care about the label model or its predicted soft labels, as long as the trained end-model is strong. Therefore, it seems helpful to **directly optimize for a strong end-model performance** rather than, e.g., maximum likelihood estimation of intermediate accuracy parameters.
> - The benefit of joint optimization over two-step approaches has been widely recognized in the crowdsourcing literature, see e.g. [1] for an early, if not the first, such approach and our paper’s references [33, 22, 40, 27, 35, 9] (we will add [1] to our references).
>
> ### Computational Efficiency
>
> We agree that computational efficiency is not the main shortcoming of the prior work, and will usually not be the main morivation for using E2E.
> However, it is worth noting that
> 1. The official implementation of Triplet(-mean) becomes quite slow for a high number of Lfs (see below for 200 LFs)
> 2. The open-source implementation of Snorkel assumes a conditionally independent model that allows them to circumvent the issue of having to estimate the inverse of the LF covariance matrix (at the cost of this Naive Bayes equivalent assumption).
>
> That being said, we are running controlled experiments where we compute the total time (in sec.) needed by E2E and baselines (training E2E/label-model + end-model+inference) for different numbers of LFs and training points.
> We will add the final results to the appendix.
>
> Here we already show an excerpt for $m=200$ LFs and n \in {10,000, 100,000} training points. E2E is significantly faster than Triplet-mean, but also significantly slower than Snorkel. (for <=100 LFs E2E is usually the slowest).
>
> $n=10,000$
>
> 	E2E     24.810   Triplet-Mean  53.487   Snorkel   11.655
>
> $n=100,000$
>
>   	E2E     185.693  Triplet-Mean  275.283  Snorkel   62.282
>
>
>
> ## Response to minor comments
> 1. **Reporting the performance of downstream or label model:** Yes, very good point! We seem to have missed making this important detail explicit somewhere that all reported metrics come are computed on the end-model -- we will add it!
> (*Note: a trained downstream model is usually the end goal of a WS pipeline, given that it can generalize to data where the LFs all abstain or are too costly to apply to by learning from informative data features. For this reason it also usually performs better than the label model, which is the case for all baseline label models in all our experiments too).*
>
> 2. **Data splits/Line 195-198:** Thanks for the feedback! The dataset splits and number of unlabeled (N_train) and labeled (N_val + N_test) data are shown in Table 3. We will rewrite this paragraph and add a pointer to the table.
>
> 3. **Line 276---what is 2b? Do you mean Figure 2b?:**
> Yes! Thank you, we have fixed this!
>
>
>
> [1] Raykar, Vikas C., et al. "Learning from crowds." Journal of Machine Learning Research (JMLR) 11.4 (2010).

---

> > ### Author Response · Authors · 2021-08-11
> > **Minor additional comment**
> >
> > We would like to briefly add with respect to
> > ``"the authors suggest their model may be performing better because the assumptions in the standard weak supervision models don't hold."``, that one of our robustness experiments (Appendix F2.2. and Figure 4) **does not violate any independence assumptions of the related work**. Even then, when adding more and more randomly voting, but independent, LFs, Snorkel and Triplets rapidly break, while E2E does so only significantly later (Fig 4).

---

### Official Review · Reviewer_DPB3 · 2021-07-17

**Rating:** 5
**Confidence:** 5

**Summary:**

This paper proposes a new method for weak supervision with multiple noisy labeling functions. The method jointly trains an encoder network that can estimate sample-by-sample accuracies with the final downstream model, and maximizes the agreement between the two models. The authors show improved performance on several weak supervision data sets and further show that their method is robust against both adversarial labeling functions and highly-correlated labeling functions.

**Limitations And Societal Impact:**

Would appreciate moving discussion of societal impacts from appendix G into the body, given the extra page provided for it in NeurIPS this year

**Main Review:**

This paper presents an interesting idea for an important problem space. The method has several benefits over previous weak supervision pipelines, in particular robustness against previous failure modes. The algorithm and paper are clearly-written, and ablations are extensive.

There are a few ways that the paper and presentation could be stronger:

1. (high-order bit) I think the paper would benefit from further analysis (not necessarily theoretical) about why the method is providing lift, especially in the challenging adversarial settings. Section 6 has some paragraphs on "why it works," but the text almost exclusively discusses degenerate solutions. This section would benefit from a more detailed analysis of which data points the method shows improvement on, and why. A few questions to ask to help get at this:
* How accurate are the soft labels produced by the model ("label model accuracy")?
* Which points do you get better soft labels for?
* How did the labeling functions vote on those points?
* Are there other points where the labeling function voted similarly, but you saw better soft labels *because* of the encoder network?

2. (more minor) I feel that the explanation for why the method avoids degenerate solutions is a little unsatisfying. The authors write that they can avoid degenerate solutions either by early stopping or by setting the temperature parameter to a magic number (< 1/3). This is promising empirically, but leaves a pit in my stomach -- how do we know that we won't see degenerate solutions on another data set? Is there any stronger guarantee that can be made to avoid degenerate solutions? Can the method be adapted at all to detect degenerate solutions and avoid them?

3. (more minor) I would be curious to see a comparison against (https://arxiv.org/pdf/2006.15168.pdf), which was a follow up to some of the baselines in the paper, and also used an encoder network to affect the soft labels from the label model. Their reported performance approaches the performance of E2E on the spouses data set (although they do not report numbers for the other data sets).

Nit: There is something wrong with the formatting of the paper -- it appears that there are almost no line breaks after paragraphs starting in section 3.4 onwards.

If these comments are addressed well (particularly 1), I think this paper could definitely move above the acceptance threshold!

Rating: 5

**Time Spent Reviewing:**

3

---

> ### Author Response · Authors · 2021-08-11
> **Response #1 to Reviewer DPB3**
>
> Thank you for your thoughtful comments and questions, which we respond to below.
>
> ### 1. Why does E2E outperform the baselines?
> Thanks for these great suggestions!
> We are actively working on a comprehensive answer to why exactly our method is able to outperform the baselines, other than the high-level intuitions regarding dropping PGM assumptions and having a more flexible and expressive model.
> As of now we do not have a concrete satisfying explanation.
>
> We would like to briefly note that E2E does not really have a label model in the sense of PGM-based methods, where you train the framework, take the label model, and use it to infer soft labels that are treated as ground truth for end-model training. Thus, *trained end-models from prior work like Snorkel are a direct result of those inferred soft labels*.
>
> In contrast, *E2E's trained end-model cannot be tracked back to a single set of soft labels or accuracy scores*. Rather, E2E's end-model is the product of complex learning dynamics and interactions with the encoder network.
> For those reasons, it is difficult to define what the "label model accuracy" of E2E is. In fact, we *do not observe soft label/label model accuracy and end-model performance to be correlated*. That is, E2E’s encoder/label model may perform relatively poorly in an epoch (e.g. worse than Snorkel’s final label model, as evaluated by the AUC to the training labels, which are otherwise unused), while in the same, or one of the next, epoch(s) the end-model reaches peak performance.
>
> This is possibly due to complex learning dynamics caused by the coupling of two distinct models, similarly to GANs.
>
> We also note that the learning scheme of our method is fairly unique/non-standard amongst deep learning models, given that it follows an unsupervised two-player agreement maximization process. It might be best comparable to non-contrastive self-supervised methods, for whose impressive empirical successes initially little explanations existed (which have now started to be given in recent follow-up papers like [1]).
>
> ### 2. Degenerate solutions
> - An extreme degenerate solution manifestates by the model’s agreeing on a single meaningless soft label for all data points (e.g. [0.5, 0.5] in the binary case). In practice, this extreme is not reached abruptly. Instead, the spread/variance of the predicted distribution decreases over training. This can be easily measured -- even without any labeled data at all -- with e.g. the standard deviation of the predicted distribution across the batch.
> However, we haven’t found a magical threshold for such a measure that allows us to rule out early on a model as being collapsed.
> - We have had successes in completely avoiding degenerate solutions via additional loss terms that enforce spread out output/softmax distributions. So far this comes at a cost in terms of end-model performance (by around 1-4 F1 points), and we are still in active research here.
>
> ### 3. Comparison to Epoxy
> Thank you for the pointer! The mentioned paper can be used complementarily to our presented method. Concretely, they use a form of LF label propagation based on pre-trained embeddings. That is, their method simply increases the coverage of existing LFs. Thus the new, extended LFs returned by their method can be used, as any other LF set, to train an arbitrary weak supervision (WS) model like ours. To draw a final important difference between this and our paper, we note that we propose a novel approach to the core WS modeling step, i.e. how to learn an end-model from multiple LFs, while they propose a method to increase LF coverage.
>
>
> ---------
>
> We will fix the line breaks between paragraphs, thanks for pointing it out!
>
>
> [1] Yuandong Tian, Xinlei Chen, Surya Ganguli:
> Understanding self-supervised learning dynamics without contrastive pairs. ICML 2021

---

### Official Review · Reviewer_SW1L · 2021-07-22

**Rating:** 5
**Confidence:** 3

**Summary:**

In this paper, the authors propose an end-to-end approach for directly learning downstream models under a weak supervision setting by maximizing its agreement with probabilistic labels generated by reparameterizing previous probabilistic posteriors with a Neural Network. Experiments show good improvement over previous works over a number of datasets.

**Limitations And Societal Impact:**

The authors briefly mentioned the limitations in the papers and my concerns are presented above.

**Main Review:**

Despite the claimed originality, similar to the provided reviews from ICML 2021, I find this paper is still a little hard to follow.
1. The introduction could be better written to make it easier for readers outside the area to follow.
2. from the problem setup in Line 95-Line 107, it seems the labeling functions just an ensemble of models performing the downstream task. Because it seems that all models in the $\mathbf{\lambda}$ perform the same $C$-way classification. This part needs to be defined more clear.
3. How specifically did the authors compute $P(y=c)$?
4. Figure 1 should have better quality.
5. I also hope that the authors could clarify more on the uniqueness and novelty of their method.
6. I'm eager to increase my score if the questions are well answered.

**Time Spent Reviewing:**

1.5 hours

---

> ### Author Response · Authors · 2021-08-11
> **Response #1 to Reviewer SW1L**
>
> First of all, we would like to thank you for the useful feedback!
> We respond below to your helpful comments and questions:
>
> ### 1. Improving the introduction
> Thanks for this feedback! If we added an example for an exemplary problem setup and LFs, would that be helpful? Are there any sentences/paragraphs that are particularly difficult to follow?
> ### 2. Labeling functions as a weak ensemble
> Indeed, we can view the set of labeling functions as an ensemble of weak models, thus the name multi-source weak supervision. *However*, the unique challenge in this setting is that the LFs may abstain ($\lambda_i = 0$ in our notation), i.e. they do not apply to all data points. This makes it often undesirable to use the LFs (or a label model) for a downstream application. For this reason we instead resort to learning an end-model that applies to arbitrary new data points based on the data features only.
>
> ### 3.  Class balance
> None of our experiments used the class balance P(Y=c), see line 125, for E2E. We did provide it to the baselines, as is usual. For the baselines, P(Y=c) was computed on the small validation set. When P(Y=c) can’t be computed, we simply assume an uniform prior.
>
> ### 4. Figure 1
>  Again, thanks for pointing this out, we will try to create a better one with draw.io or TikZ (instead of google drawings).
>
> ### 5. Uniqueness and Novelty of E2E
> - We note that our method is to the best of our knowledge the only existing alternative to the prevalent PGM-based methods (i.e. snorkel and triplet-methods) that applies to the standard multi-source weak supervision (MSWS) setting in its full generality.
> With standard MSWS setting we refer to the (multi-class) setup introduced by the seminal data programming paper [1]. Notably, this does not assume access to any labeled data.
> Given that our approach is based on a (novel) neural network parameterization, and thus is in stark contrast to PGM-based methods, we believe that our work is quite novel.
> - It is also worth mentioning that amongst weak supervision papers, our method has unique parallels to the recent, state-of-the-art literature on self-supervised learning (SSL, see e.g.: [2, 3, 4]) that may open the path for cross-pollination between MSWS and SSL.
> Interesting parallels are 1. the use of a pair of neural nets that are trained on each others predictions; 2. the existence of degenerate solutions due to the missing (strong) supervision via ground truth labels; and 3. the importance of the Stop-Grad operation (for non-contrastive SSL [3, 4])
>
> [1] Alexander Ratner, Christopher De Sa, Sen Wu, Daniel Selsam, and Christopher Ré. 2016. Data programming: creating large training sets, quickly. In Proceedings of the 30th International Conference on Neural Information Processing Systems (NeurIPS'16).
>
> [2] Chen, Ting, et al. "A simple framework for contrastive learning of visual representations." International conference on machine learning (ICML), 2020.
>
> [3] Grill, Jean-Bastien, et al. "Bootstrap your own latent: A new approach to self-supervised learning." arXiv preprint arXiv:2006.07733 (2020).
>
> [4] Chen, Xinlei, and Kaiming He. "Exploring simple siamese representation learning." Proceedings of the IEEE/CVF Conference on Computer Vision and Pattern Recognition. 2021.

---

> > ### Comment · Reviewer_SW1L · 2021-09-02
> > **Update**
> >
> > Thank you for the responses to my concerns regarding the paper. After going through the rebuttal and reading the other reviews, I decided to keep my original score.

---

### Official Review · Reviewer_jomY · 2021-07-25

**Rating:** 5
**Confidence:** 4

**Summary:**

The authors propose an "end to end" approach to weakly supervised learning. Instead of obtaining ground truth labels by modeling WS sources via a probabilistic latent variable model which don't consider the downstream performance, the authors consider a neural model that directly learning the downstream model by maximizing WS sources' agreement with probabilistic labels generated by reparameterizing previous probabilistic posteriors. The method outperforms prior approaches on five real-world datasets.

**Limitations And Societal Impact:**

The authors discussed their limitations on probabilistic labeling functions and modeling more structure.


**Main Review:**

- Originality: The method proposed in this work is new. Most of the existing work in weakly supervised learning focuses on only create accurate training labels via WS resources. Constrastingly, this work focuses on developing a "end to end" approach by condiering downstream performance.

- Quality: The paper has overall good experimental results while some strong claims lack of theoretical or empirical support. For example, 1) The ablation section is weak. An ablation study is needed to show in which case or which condition the proposed approach outperform the prior works. 2) It would be great to show whether the proposed approach works for different types of models, e.g., transformer-based model.

- Clarity: The writing is overall clear. While the experiment section is difficult to understand because of lack of organization, or flaws.

- Significance: The key insight of the paper that weakly supervised learning can be benefit by direcly considering downstream performance. This paper proposes a end-to-end approach to weak supervision modeling. While overall novaltiy is minor.

======== After rebuttal ========

Thank you for the detailed response. My questions are partially answered by the authors, and I would like to keep my original rating.



**Time Spent Reviewing:**

4

---

> ### Author Response · Authors · 2021-08-11
> **Response #1 to Reviewer jomY**
>
> First of all, we would like to thank you for the useful feedback and comments, to which we respond below.
>
> ### Ablation sections
> Besides the comprehensive model design ablation in appendix D, we provided ablations for 3 different conditions, which indicate that our method is more robust than  prior work (i.e.outperforms it) under the presence of:
> 1. (strong) correlations amongst the LFs (Fig 2b. and appendix F.2.1)
> 2.  adversarial LFs (Fig 2a. and section 4.3),
> 3.  label noise from *independent*, no-better-than-random-LFs (appendix F.2.2).
> Notably, this condition does not break the conditionally independent/Naive Bayes assumption of the related work.
>
> This is complemented by the main experiments on 5 weak supervision (WS)
> datasets and 6 LF settings, including a crowd-sourcing dataset, where we aim to show that our method also performs comparably or better under the diverse WS conditions found in these datasets.
>
> ### Other end-models/Transformers
> We agree that it is important for our method to work with different types of (end-)models. In the present draft, our experiments already include various MLPs, a LSTM, as well as fine-tuning features extracted from a pre-trained CNN (a VGG) in the crowd-sourcing experiment.
> While a Transformer would be interesting too, training it from scratch would first require the creation of a WS dataset with an appropriate sample size.
>
>
> ### Comments on:
> - **Experimental flaws**: Any pointers or concrete feedback to present flaws in our experiments would be very much appreciated so that we can fix them and improve our manuscript.
>
> - **Limitations**: We would like to note that both  probabilistic labeling functions (PLFs) and "modeling more structure" are not limitations of our proposed method but rather straight-forward extensions to it that distinguish it from prior work (e.g., none of the baselines are able to support PLFs).

---

### Decision · Program_Chairs · 2021-09-27

**Decision:**

Accept (Poster)

**Comment:**

This paper proposes a deep learning, end-to-end approach to learning from multiple noisy labeling functions. The proposed method outperforms existing techniques that use probabilistic graphical models for label aggregation. One of the major criticisms of the paper is that the paper does not analyze (theoretically or empirically) why exactly the proposed technique works well. The reviews believe that they have suggested different empirical investigations, but the authors did not engage with these ideas during the rebuttal period. The reviewers were willing to raise their rating if additional empirical investigation about the inner-workings of the technique had been conducted. Other that the lack of analysis, the reviewers do not present a strong criticism of the technique and are happy with empirical results. I do resonate with the authors that deep learning techniques can be challenging to analyze, and my overall reasoning is in agreement with reviewer #zED9 that the novelty of the proposed method along with the improved empirical performance can substitute the lack of theoretical or empirical justification to some extent. Overall, I believe the publication of this paper can help the research community make progress on finding better algorithms for aggregating multiple noisy labeling sources, and better understanding of this algorithm can be left for future work. Hence, I recommend acceptance as a poster.